# Health literacy assessment and healthcare access difficulties of Vietnamese migrants in Japan: A cross-sectional study

Takashi Tsubakita[1], Nobuo Kawazoe[2]*, Nobuyuki Matsuo[2]

1 Department of Management, Nagoya University of Commerce and Business, Nisshin, Aichi, Japan,
2 Department of Economics, Nagoya University of Commerce and Business, Nisshin, Aichi, Japan

* kawazoe@nucba.ac.jp

## Abstract

While the number of Vietnamese migrant workers in Japan has been increasing, their healthy literacy as a key concern for ensuring equitable access to healthcare is unknown. This study assessed health literacy among Vietnamese migrants in Japan and examined their access to healthcare and the difficulties they encounter. Convenience sampling was employed, with Vietnamese support organizations across Japan invited to disseminate a web-based survey via social media platforms. Using both self-reported and test-based health literacy tools, we measured health literacy levels in a sample of 137 Vietnamese migrants. We identified disparities by gender and topic area, where women were likely to score higher than men, particularly in mental health and knowledge of sexually transmitted infections. Our findings underscore the importance of culturally tailored health education and community-based interventions to support the health of this population. The study highlights the need for cultural and linguistically adapted educational materials to improve equitable access to healthcare. Our work contributes to the ongoing dialogue on migrant health and the development of inclusive public health strategies.

## Introduction

In 2022, the World Health Organization highlighted critical health disparities among the world's 280 million migrants, particularly low-skilled workers who face limited access to healthcare and an increased risk of mental health issues such as depression and post-traumatic stress disorder [1]. Sustainable Development Goal 3 of the UN's 2030 Agenda calls for universal health coverage, emphasizing equitable access to essential health services and affordable medicines [2]. Addressing migrant health requires comprehensive strategies, including enhancing physical and financial access, improving health education, ensuring culturally and linguistically appropriate care, and strengthening healthcare infrastructure [2].

**Data availability statement:** All relevant data are within the paper and its Supporting Information files.

**Funding:** TT. This study was supported by a grant from the Daiko Foundation of Japan.

**Competing interests:** The authors have declared that no competing interests exist.

As of 2023, Japan had 3.13 million foreign residents, representing a 10.9% increase from the previous year [3]. Vietnamese nationals constitute the second-largest foreign population in Japan. Despite this growth, language barriers and limited understanding of Japan's healthcare system continue to hinder access to healthcare services [2]. Although some support systems, such as multilingual materials, legal consultations, and migrant-friendly directories, have been introduced, navigating these systems still relies heavily on individuals or their employers [4]. Community-based support often remains limited to native-language networks, underscoring the need for broader access to health information, interpreter services, and culturally sensitive care [5,6]. Local Vietnamese associations and public agencies have begun addressing these issues through health booklets and community outreach.

To promote healthier lives for migrants in Japan, it is essential to improve their health literacy, defined as the ability to understand health systems, access accurate information, and appropriately utilize healthcare [7,8]. Low levels of health literacy are consistently associated with poorer health outcomes, including reduced use of preventive care, higher hospitalization rates, and weaker self-management of chronic conditions [9]. To overcome these challenges, support from local communities is vital to help individuals navigate the health system and access the information they need [10]. Evidence suggests that, in rural areas of Vietnam, community dynamics play a substantial role in shaping individuals' health literacy [11]. In Vietnam, health literacy research often employs the 47-item European Health Literacy Questionnaire (HLS-EU-Q47), which is closely linked to education and socioeconomic status [12,13]. Outside Vietnam, studies indicate that language and cultural barriers continue to restrict access to health information, particularly for women and immigrants in countries such as Australia and the United States [14,15]. To the best of our knowledge, however, no prior study has specifically investigated health literacy among Vietnamese migrants in Japan.

### Objectives

This study quantitatively assessed health literacy among Vietnamese migrants in Japan and examined their access to healthcare as well as the difficulties they encounter.

## Materials and methods

### Participants

According to the Ministry of Health, Labor, and Welfare of Japan, there were 1.82 million foreign workers in Japan in 2022, of whom approximately 460,000 (25%) were Vietnamese. Based on the prior studies [14–15], this survey focused on Vietnamese workers likely to face greater challenges in health literacy: those employed in whole-sale and retail, food services and accommodation, construction, manufacturing, and agriculture-related sectors. The survey also included Vietnamese individuals living in Japan as students or family members. Convenience sampling was employed. Vietnamese support organizations across Japan were asked to disseminate a web-based survey through social media platforms. Participants were informed that participation

was voluntary, that their data would remain confidential, and that they could withdraw from the study at any time. The participants recruitment period started on July 28, 2024, and ended on January 16, 2025. Only responses from individuals who provided informed consent in an implied manner were included in the analysis. This study was approved by the Ethics Committee of Nagoya University of Commerce and Business (Approval No. 23034).

### Questionnaire

The questionnaire included demographic and health-related items, such as age, sex, history of illness or injury while living in Japan, whether medical care was sought, difficulties encountered during care, preferences for Vietnamese versus Japanese medications, and perceived barriers to accessing services.

The 12-item Short-Form Health Literacy Questionnaire (HLS-SF12) [12], adapted from the HLS-EU-Q47 [16], is widely used in Asia. It assesses perceived difficulty with health-related tasks across three domains: healthcare (items 1–4), disease prevention (items 5–8), and health promotion (items 9–12). Items are rated on a four-point Likert scale ranging from "very difficult" (1 point) to "very easy" (4 points). The functional Health Literacy Scale (funHLS) [17] is a 25-item multiple-choice test measuring basic health knowledge regarding the body, illness, nutrition, and medical care. Each item presents a health-related term and asks respondents to select the most relevant option or choose "I don't know." Higher scores on this test indicate greater functional knowledge.

### Data analysis

The number of respondents who reported illness or injury and their healthcare-seeking behaviors were recorded. Means and standard deviations were calculated for the three HLS-SF12 domains. For the funHLS, the correct answer rate for each item and the overall mean score were computed. A goodness-of-fit test was used to compare men and women in Table 7. All analyses were performed using Stata version 19.

### Results

In total, 141 responses were received from Vietnamese migrants residing in Japan. Of these, 137 responses were included in the final analysis after excluding four with substantial missing data. Table 1 summarizes the demographic characteristics of the participants. The sample consisted of 50 men (36.5%) and 87 women (63.5%). The mean age was 29.1 years, with a median of 28.0 years. The largest age group was 25–29 years, followed by 20–24 and 30–34 years. In terms of education, 44.5% had graduated from university, 42.3% from high school, and 8.0% from graduate school. Only a few participants reported completing junior high school or other forms of education.

Table 2 presents employment sectors, visa status, length of stay, and region of residence. Participants were engaged in diverse sectors, with machinery manufacturing (20.4%) and food manufacturing (19.7%) being the most common, together accounting for approximately 40% of the sample. Other sectors included construction, textile manufacturing, agriculture, and fisheries. A large proportion (43.1%) were categorized as "other," reflecting a variety of occupations.

Regarding visa status, most participants were either workers (43.1%) or "admitted trainees" (29.9%). The admitted trainee program refers to workers admitted under a government scheme allowing limited-term employment in Japan. Family members, and permanent residents, and students, accounted for 10.9%, 8.0%, and 7.3%, respectively. Only one participant was classified as "other," suggesting that most respondents held work-related visas.

In terms of length of stay, 36.5% had lived in Japan for 1–2 years, while 19.0%, 21.9%, and 17.5% had stayed for 3–4, 5–6, and 7 years or more, respectively. Only 5.1% had been in Japan for less than one year. The mean and median durations were 3.8 and 4.0 years, respectively.

Regarding region of residence, 48.9% lived in western Japan, followed by 25.5% in northern regions. Smaller proportions resided in central, eastern, and southern Japan.

**Table 1. Participants' demographic characteristics.**

| Variables | | n (%) |
|---|---|---|
| Sex | All | 137 (100%) |
| | Men | 50 (36.5) |
| | Women | 87 (63.5) |
| Age group, years | 17-19 | 2 (1.5) |
| | 20-24 | 30 (21.9) |
| | 25-29 | 49 (35.8) |
| | 30-34 | 30 (21.9) |
| | 35-39 | 15 (10.9) |
| | 40-44 | 11 (8.0) |
| | Mean 29.1 | |
| | Median 28.0 | |
| Education | Junior high school | 5 (3.6) |
| | High school | 58 (42.3) |
| | University | 61 (44.5) |
| | Graduate school | 11 (8.0) |
| | Others | 2 (1.6) |

**Table 2. Participants' living status.**

| Variables | | n (%) |
|---|---|---|
| Employment sector | Machinery manufacturing | 28 (20.4) |
| | Food manufacturing | 27 (19.7) |
| | Construction | 9 (6.6) |
| | Textile manufacturing | 8 (5.8) |
| | Agriculture | 4 (2.9) |
| | Fisheries | 2 (1.5) |
| | Others | 59 (43.1) |
| Visa status | Workers | 59 (43.1) |
| | Admitted trainees | 41 (29.9) |
| | Family members | 15 (10.9) |
| | Permanent residents | 11 (8.0) |
| | Students | 10 (7.3) |
| | Others | 1 (0.7) |
| Length of stay, years | < 1 | 7 (5.1) |
| | 1-2 | 50 (36.5) |
| | 3-4 | 26 (19.0) |
| | 5-6 | 30 (21.9) |
| | 7- | 24 (17.5) |
| | Mean 3.8 | |
| | Median 4.0 | |
| Region of residence in Japan | North | 35 (25.5) |
| | East | 8 (5.8) |
| | Middle | 25 (18.2) |
| | West | 67 (48.9) |
| | South | 2 (1.5) |

## Access to healthcare and related challenges

Among the 137 participants, 77 reported seeking medical care for illnesses or injuries since arriving in Japan. The main difficulties encountered included communication barriers (70.1%), long waiting times (59.7%), not knowing which hospital to visit (53.2%), and high fees (50.6%). Additional issues included expensive medicines, limited understanding of the healthcare system, and uncertainty about where to purchase medicines (Table 3).

In contrast, 27 participants reported experiencing serious illness or injury but did not seek medical care. Reasons included not knowing which hospital department to visit (70.4%), perceiving the condition as not severe enough (66.7%), difficulty communicating in Japanese (59.3%), and, to a lesser extent, lack of time, high costs, long waiting times, inconvenient transportation, and absence of nearby hospitals (Table 4).

When asked about medication preferences, 78 participants reported using Japanese medicines. Reasons included trust of Japanese medication in quality (89.7%), Familiarity (44.9%), and, to a lesser extent, the availability of medicines not found in Vietnam and familiarity (Table 5).

## Self-reported measures

Table 6 shows participants' self-reported scores on the HLS-SF12 across three domains.

For the healthcare domain, the mean score (M) was 9.12, with no notable sex difference (men: 9.04; women: 9.16). Tasks such as "finding information on diseases" (M = 2.42) and "calling an ambulance" (M = 2.27) were rated as easier, while "judging pros and cons of multiple treatment options" (M = 2.10) was considered most difficult. For the disease

**Table 3. Difficulties in doctor visits (multiple answers).**

| Difficulties# | n (%) |
|---|---|
| Total | 77 (100%) |
| Communication | 54 (70.1) |
| Long waiting time | 46 (59.7) |
| Hospital selection | 41 (53.2) |
| Expensive fee | 39 (50.6) |
| Expensive medicines | 33 (42.9) |
| Understanding healthcare system | 25 (32.5) |
| Where to get medicines | 23 (29.9) |

#A participant answered multiple difficulties in this table.

**Table 4. Reasons of avoiding visiting doctors when illness or injured.**

| Reasons# | n (%) |
|---|---|
| Total | 27 (100%) |
| No knowledge about proper doctors | 19 (70.4) |
| Seemed not so severe enough | 18 (66.7) |
| Difficulties in Japanese communication | 16 (59.3) |
| Lack of time | 15 (55.6) |
| High costs | 15 (55.6) |
| Long waiting time | 15 (55.6) |
| Inconvenient transportation | 8 (29.6) |
| Absence of nearby hospitals | 5 (18.5) |

#A participant answered multiple reasons in this table.

**Table 5. Reasons of taking Japanese medicines instead of Vietnamese ones.**

| Reasons# | n (%) |
|---|---|
| Total | 78 (100%) |
| Trust in quality | 70 (89.7) |
| Familiarity | 35 (44.9) |
| Not found in Vietnam | 29 (37.2) |
| Lower costs | 12 (15.4) |

#A participant answered multiple reasons in this table.

**Table 6. Self-reported scores based on HLS-SF12.**

| | All, n=137 | Men, n=50 | Women, n=87 |
|---|---|---|---|
| Questions | Mean (SD) | Mean (SD) | Mean (SD) |
| **Health care** | 9.12 (2.24) | 9.04 (1.99) | 9.16 (2.52) |
| Q1. Informaton on deseases | 2.42 (0.73) | 2.34 (0.66) | 2.46 (0.77) |
| Q2. Understanding leaflets | 2.33 (0.75) | 2.32 (0.62) | 2.33 (0.82) |
| Q3. Judgement the advantages of treatments | 2.10 (0.70) | 2.10 (0.68) | 2.10 (0.72) |
| Q4. Amburance call | 2.27 (0.81) | 2.28 (0.76) | 2.26 (0.84) |
| **Preventive medicine** | 10.03 (2.48) | 9.62 (2.36) | 10.26 (2.52) |
| Q5. Finding informaton on deseases | 2.35(0.76) | 2.32 (0.68) | 2.37 (0.81) |
| Q6. Understanding health screenings | 2.52 (0.73) | 2.44 (0.67) | 2.56 (0.76) |
| Q7#. Judgement on the vaccination | 2.49 (0.75) | 2.30 (0.74) | 2.60 (0.76) |
| Q8. Judgement on the treatment | 2.67 (0.76) | 2.56 (0.76) | 2.74 (0.75) |
| **Health promotion** | 10.72 (2.39) | 10.76 (2.40) | 10.70 (2.39) |
| Q9. Contact with health activities | 2.80 (0.71) | 2.84 (0.74) | 2.77 (0.69) |
| Q10. Understanding on media information | 2.80 (0.75) | 2.84 (0.77) | 2.78 (0.74) |
| Q11. Judgement on the healthy life style | 2.79 (0.72) | 2.76 (0.69) | 2.80 (0.74) |
| Q12. Participating physical training activities | 2.34 (0.81) | 2.32 (0.82) | 2.34 (0.80) |

SD, Standard deviation

#Men <Women (P value = 0.025)

The Q1-Q12 below are questions in Table 6.

Q1. Do you find information on treatments of illnesses that concern you?

Q2. Do you understand the leaflets that come with your medicine?

Q3. Do you judge the advantages and disadvantages of different treatment options?

Q4. Do you call an ambulance in an emergency?

Q5. Do you find information on how to manage mental health problems like stress or depression?

Q6. Do you understand why you need health screenings (such as breast exam, blood sugar test, blood pressure)?

Q7. Do you judge which vaccinations you may need?

Q8. Do you decide how you can protect yourself from illness based on advice from family and friends?

Q9. Do you find out about activities (such as meditation, exercise, walking, Pilates etc.) that are good for your mental well-being?

Q10. Do you understand information in the media (such as Internet, newspaper, magazines) on how to get healthier?

Q11. Do you judge which everyday behavior (such as drinking and eating habits, exercise etc.) is related to your health?

Q12. Do you join a sports club or exercise class if you want to?

prevention domain, the mean score was 10.03, with women scoring higher than men (10.26 vs. 9.62). Participants found "understanding the need for health checkups" (M = 2.52) and "judging the need for vaccinations" (M = 2.49) easier, while "finding mental health information" (M = 2.35) was more difficult. A significant sex difference was observed for "judging the need for vaccinations," where women scored higher than men (2.60 vs. 2.30, p = 0.025).

For the health promotion domain, the mean score was 10.72, with no sex differences. Respondents rated "understanding health media information" (M = 2.80) and "judging healthy lifestyles" (M = 2.79) as easier. However, "participating in physical training activities" scored lower (M = 2.34), suggesting greater difficulty.

### Test-based measures

The funHLS is a measurement tool of health literacy based on tests. Table 7 presents item-wise correct response rates for the 25 items.

Overall, correct response rates varied widely. High accuracy was observed for "syphilis" (83%), "health insurance card" (82%), "kidney" (77%), and "pneumothorax" (76%). Low rates were recorded for "generic drugs" (6%), "amenorrhea" (18%), "sexually transmitted infections" (23%), and "autism spectrum disorder" (28%).

**Table 7. Test-based scores based on funHLS.**

| | All, n = 137 | Men, n = 50 | Women, n = 87 | p value |
|---|---|---|---|---|
| Question items | Ratio (SD) | Ratio (SD) | Ratio (SD) | |
| Q1. Caries | 0.74 (0.44) | 0.76 (0.43) | 0.74 (0.44) | 0.753 |
| Q2. Prescription | 0.61 (0.49) | 0.48 (0.50) | 0.68 (0.47) | 0.022* |
| Q3. Generic drug | 0.06 (0.24) | 0.06 (0.24) | 0.06 (0.23) | 0.952 |
| Q4. Measles | 0.52 (0.50) | 0.44 (0.50) | 0.56 (0.50) | 0.165 |
| Q5. Sexually transmitted infections | 0.23 (0.42) | 0.14 (0.35) | 0.28 (0.45) | 0.067 |
| Q6. Anemia | 0.76 (0.43) | 0.68 (0.47) | 0.80 (0.40) | 0.101 |
| Q7. Potassium | 0.56 (0.50) | 0.54 (0.50) | 0.57 (0.50) | 0.693 |
| Q8. AED | 0.28 (0.45) | 0.30 (0.46) | 0.28 (0.45) | 0.763 |
| Q9. Fat | 0.53 (0.50) | 0.46 (0.50) | 0.57 (0.50) | 0.195 |
| Q10. Depression | 0.55 (0.50) | 0.46 (0.50) | 0.61 (0.49) | 0.091 |
| Q11. Amenorrhea | 0.18 (0.38) | 0.08 (0.27) | 0.23 (0.42) | 0.026* |
| Q12. Schizophrenia | 0.65 (0.48) | 0.58 (0.50) | 0.69 (0.47) | 0.195 |
| Q13. Autism spectrum disorder | 0.28 (0.45) | 0.28 (0.45) | 0.29 (0.46) | 0.927 |
| Q14. Revisit to clinic | 0.55 (0.50) | 0.60 (0.49) | 0.52 (0.50) | 0.349 |
| Q15. Kidney | 0.77 (0.42) | 0.72 (0.45) | 0.80 (0.40) | 0.255 |
| Q16. Pneumothorax | 0.76 (0.43) | 0.76 (0.43) | 0.76 (0.43) | 0.985 |
| Q17. Medical treatment not coverd by insurance | 0.46 (0.50) | 0.36 (0.48) | 0.52 (0.50) | 0.075 |
| Q18. Carbohydrates | 0.57 (0.50) | 0.54 (0.50) | 0.59 (0.50) | 0.599 |
| Q19. BMI | 0.50 (0.50) | 0.48 (0.50) | 0.52 (0.50) | 0.675 |
| Q20. Salt | 0.68 (0.47) | 0.62 (0.49) | 0.71 (0.45) | 0.264 |
| Q21. Insurance card | 0.82 (0.38) | 0.84 (0.37) | 0.82 (0.39) | 0.723 |
| Q22. Appendicitis | 0.76 (0.43) | 0.72 (0.45) | 0.78 (0.42) | 0.417 |
| Q23. Vitamin C | 0.63 (0.49) | 0.54 (0.50) | 0.68 (0.74) | 0.107 |
| Q24. Syphilis | 0.83 (0.38) | 0.78 (0.42) | 0.86 (0.35) | 0.216 |
| Q25. Uterocervical cancer | 0.31 (0.46) | 0.26 (0.44) | 0.33 (0.47) | 0.370 |

SD, Standard deviation. The p-value for sex-based comparison for each item is shown. *p < 0.05

Sex-based comparisons showed that women scored higher than men overall. Significant differences were found for "prescription" (68% vs. 48%, p = 0.022) and "amenorrhea" (23% vs. 8%, p = 0.026), where women outperformed men. None of the items showed significantly higher scores among men. By category, knowledge of infectious diseases and vaccinations varied: correct response rates were 52% for "measles," 23% for "sexually transmitted infections," and 83% for "syphilis," reflecting relatively low awareness of sexually transmitted infections. Regarding mental and neurological health, correct response rates for "schizophrenia" (65%), "depression" (55%), and "autism spectrum disorder" (28%) showed that while some conditions were better understood, developmental disorders were less recognized.

Knowledge of nutrition and lifestyle was relatively stable, with moderate accuracy for "salt" (68%), "carbohydrates" (57%), and "BMI" (50%). However, responses for specific nutrients such as "potassium" (56%) varied, suggesting uneven understanding in this area.

## Discussion

### Access to healthcare and related challenges

Many Vietnamese migrants in Japan experienced illnesses or injuries and sought medical care. However, a notable proportion did not access medical services even when needed. The contributing factors included limited time due to work, high medical costs, and language barriers. Several participants reported confusion about which hospital to visit or how the healthcare system operates, indicating insufficient knowledge of Japan's medical infrastructure. Communication difficulties with medical staff were also common and may have hindered appropriate treatment. Long waiting times and high fees further discouraged access to care. Some participants avoided medical services because of difficulties navigating the system in Japanese or uncertainty about the appropriate department to visit. In some cases, geographic isolation and inadequate transportation also limited access.

Regarding medications, both Japanese and Vietnamese medicines were used. Japanese medicines were preferred for their reliability and affordability, while Vietnamese medicines were chosen due to familiarity and the availability of certain types not found in Japan. This reflects a continued demand for both.

In summary, although many Vietnamese migrants in Japan use healthcare services, access is often restricted by systemic, linguistic, and financial challenges. To improve health equity, support through accessible medical information, stronger language services, and reduced financial burdens is necessary.

### Health literacy from the perspective of task difficulty

In the healthcare domain of the HLS-SF12, no significant sex differences were observed. Participants found basic tasks such as finding information on illnesses or calling an ambulance relatively easy but had difficulty "judging the pros and cons of multiple treatment options." This suggests adequate competence in basic health-related tasks but persistent challenges in more complex health-related decision-making, likely due to limited access to professional information or system-level knowledge. Language barriers and a lack of health-related education may also contribute to these challenges. In the disease prevention domain, women scored higher overall, particularly on "judging the need for vaccinations." This may reflect women's greater interest in preventive health or their increased exposure to related information. Conversely, low scores for "finding mental health information" suggest limited access to such resources or high barriers to seeking them.

In the health promotion domain, the mean score was 10.72, with no sex difference. Respondents found it relatively easy to understand health-related media information and wellness concepts, such as meditation or walking. However, lower scores for "participating in physical training activities" suggest barriers to actual engagement, likely stemming from financial, time, or language constraints.

### Test-based health literacy

Results based on the funHLS varied depending on topic knowledge. Participants performed well on familiar items such as "insurance card," "syphilis," "kidney," and "pneumothorax," but poorly on "generic drugs," "amenorrhea," "sexually

transmitted infections," and "autism spectrum disorder," which require more technical knowledge. Men scored particularly low on certain items compared with women, suggesting sex-based disparities in exposure or health-related knowledge.

Notably, awareness of sexually transmitted infections was limited. It is possible that the Vietnamese term for sexually transmitted infections is not widely known or commonly used. While awareness of conditions such as measles and syphilis was higher, broader knowledge of infectious diseases was inconsistent. In mental health, participants showed moderate understanding of depression and schizophrenia but low recognition of autism spectrum disorder, suggesting insufficient awareness of developmental disorders and the need for targeted education [18].

Knowledge of basic nutrition topics such as salt, carbohydrates, and BMI was moderate, but responses regarding specific nutrients, such as potassium, were less consistent. These findings highlight gaps in access to detailed health information and uneven educational exposure.

### Perspective on health literacy

Overall, the findings reveal uneven distribution of knowledge and skills, with challenges in complex decision-making and healthcare system navigation. Women generally scored higher than men, perhaps reflecting their greater tendency to seek health-related information. These results underscore the importance of actively supporting men's health literacy as well.

Multilevel approaches are required to improve health literacy among Vietnamese migrants in Japan. These include improving access to medical information, providing language support and targeted health education, and implementing strategies to reduce financial burdens. In addition to localized or workplace-based interventions, programs focusing on mental health and STIs are especially crucial. Sustained, culturally responsive efforts are essential to enhance the well-being and healthcare navigation skills of Vietnamese migrants in Japan.

### Limitations

This study has several limitations. First, although participants were recruited from across Japan, the sample size was relatively small, and convenience sampling was used. Therefore, the findings may not be generalizable to the broader population of Vietnamese migrants in Japan. There may also be occupational biases. For example, 44.5% of respondents were university graduates, whereas the estimated proportion of university graduates in Vietnam is about 20%. Thus, the sample was skewed toward individuals with higher educational attainment. Second, the study relied on self-reported data, which are subject to recall and social desirability biases. Participants may have over- or underestimated their health behaviors or knowledge. Finally, although this study considered linguistic and cultural barriers, it did not examine in detail how these factors specifically influenced health literacy. Future research should explore the roles of language proficiency, cultural adaptation, and prior health education more thoroughly. Despite these limitations, this study represents one of the first attempts to quantitatively assess health literacy among Vietnamese migrants in Japan and provides important insights to inform policy and intervention development.

### Conclusion

This study highlights the importance of culturally and linguistically tailored educational materials in improving equitable access to healthcare. Effective measures may include interpretation services, navigation support, and targeted outreach through community and workplace channels. Strategies to better engage male migrants in health promotion are also recommended.

### Supporting information

**S1 File. Datafile.**
(XLSX)

**S2 File. Response to Reviewers.**
(DOCX)

**S3 File. PLOS ONE.**
(PDF)

## Acknowledgments

We would like to express our sincere gratitude to the Vietnamese residents in Japan who kindly participated in the survey. We also extend our appreciation to the members of Vietnamese associations who supported participant recruitment.

## Author contributions

**Conceptualization:** Takashi Tsubakita.

**Data curation:** Takashi Tsubakita, Nobuo Kawazoe, Nobuyuki Matsuo.

**Formal analysis:** Nobuo Kawazoe, Nobuyuki Matsuo.

**Investigation:** Nobuyuki Matsuo.

**Methodology:** Nobuo Kawazoe.

**Project administration:** Takashi Tsubakita, Nobuo Kawazoe.

**Writing – original draft:** Takashi Tsubakita.

**Writing – review & editing:** Nobuo Kawazoe, Nobuyuki Matsuo.

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
