## [Decision Letter · Decision Letter 0]

27 Jan 2026

Dear Dr. Kawazoe,

Thank you for submitting your manuscript to PLOS ONE. After careful consideration, we feel that it has merit but does not fully meet PLOS ONE’s publication criteria as it currently stands. Therefore, we invite you to submit a revised version of the manuscript that addresses the points raised during the review process.

**ACADEMIC EDITOR: Please insert comments here and delete this placeholder text when finished.**

Address all comments by reviewers, especial those on the attached documents from reviewer 1 and 3.

We look forward to receiving your revised manuscript.

Kind regards,

Immaculate Sabelile Tenza, PhD

Guest Editor

PLOS One

Journal Requirements:

4. In the online submission form, you indicated that the data sets are available upon request from: kawazoe@nucba.ac.jp

Additional Editor Comments :

Thank you for choosing our journal for publication

Kindly respond to the reviewers comments

Language editing is recommended to the manuscript

Reviewers' comments:

Reviewer's Responses to Questions

**Comments to the Author**

1. Is the manuscript technically sound, and do the data support the conclusions?

Reviewer #1: Yes

Reviewer #2: Yes

Reviewer #3: Yes

2. Has the statistical analysis been performed appropriately and rigorously?

Reviewer #1: Yes

Reviewer #2: Yes

Reviewer #3: N/A

3. Have the authors made all data underlying the findings in their manuscript fully available?

Reviewer #1: Yes

Reviewer #2: Yes

Reviewer #3: Yes

4. Is the manuscript presented in an intelligible fashion and written in standard English?

Reviewer #1: Yes

Reviewer #2: Yes

Reviewer #3: Yes

Reviewer #1: The manuscript has a potential to contribute to the body of knowledge. There are minor corrections as indicated in the uploaded file which, when addressed can improve the quality of the manuscript. The author is directed to the uploaded file for detailed comments

Reviewer #2: The manuscript is well written and data well presented. However, being the first study of its kind, the authors may rephrase their first sentence in their abstract to read .."their healthy literacy as a key concern for ensuring equitable access to healthcare is unknown." Further, under the section named Data analysis, the authors have stated the software used and the descriptive analyses, however, they need to specify the tests and model (s) used in the study.

Reviewer #3: Abstract: Specify the health literacy tools used (HLS-SF12, funHLS).

Methods: Clarify the recruitment window (28/07/2024 to 16/01/2025 seems unusually long or is a typo for 2023-2024?).

Ethics: The consent procedure ("consent... considered to be given by the act of completing the questionnaire") should be explicitly labeled as implied consent. in scientific research informed consent is often best practice.

References: Formatting is inconsistent (some titles capitalized, others not). Check journal style.

**Do you want your identity to be public for this peer review?** For information about this choice, including consent withdrawal, please see our Privacy Policy

Reviewer #1: **Yes:** Dr Gopolang Gause

Reviewer #2: **Yes:** Dr Patrick M. Mutua

Reviewer #3: No

---

## [Author Response · Author response to Decision Letter 1]

4 Feb 2026

Response to Review Comments to the Author

We sincerely thank the Academic Editor and three reviewers for your careful review and constructive comments on our manuscript. We greatly appreciate your insightful suggestions, which have substantially helped us improve the clarity, transparency, and overall quality of the manuscript.

Reviewer #1: The manuscript has a potential to contribute to the body of knowledge. There are minor corrections as indicated in the uploaded file which, when addressed can improve the quality of the manuscript. The author is directed to the uploaded file for detailed comments

Comment:

We sincerely thank you for your valuable comments. All our responses for your comments are written in the uploaded file.

Reviewer #2: The manuscript is well written and data well presented. However, being the first study of its kind, the authors may rephrase their first sentence in their abstract to read: "their healthy literacy as a key concern for ensuring equitable access to healthcare is unknown." Further, under the section named Data analysis, the authors have stated the software used and the descriptive analyses, however, they need to specify the tests and model (s) used in the study.

Comment:

We replaced the sentence to make the focus of the study clear. Thank you for your precise advice.

This revision can be found in the Abstract (Lines 2–3).

For the data analysis, we added the sentence "A goodness-of-fit test was used to compare men and women in Table 7”.

This revision can be found in Materials and methods (Lines 90).

Reviewer #3:

Abstract: Specify the health literacy tools used (HLS-SF12, funHLS).

Comment:

We used the HLS-SF12 as a self-reported tool and the funHLS as a test-based tool. Table 6 shows the result based on the former tool and Table 7 shows the result based on the latter.

Methods: Clarify the recruitment window (28/07/2024 to 16/01/2025 seems unusually long or is a typo for 2023-2024?).

Comment:

We replaced the date expression 28/07/2024, 16/01/2025 by July 28, 2024, January 16, 2025, respectively.

Ethics: The consent procedure ("consent... considered to be given by the act of completing the questionnaire") should be explicitly labeled as implied consent. in scientific research informed consent is often best practice.

Comment:

In the first part of questionnaire, ethical explanations were written. We asked participants to read them before answering the questionnaire and to answer it only if they got informed consent. The implied consent was provided. We rewrote this process of informed consent in detail in the Ethical statement.

References: Formatting is inconsistent (some titles capitalized, others not). Check journal style.

Comment:

We corrected formatting inconsistency of capital letters in the References according to the journal style.

---

## [Editor Report · Decision Letter 1]

25 Feb 2026

Health literacy assessment and healthcare access difficulties of Vietnamese migrants in Japan: a cross-sectional study

PONE-D-25-55029R1

Dear Dr. Kawazoe,

We’re pleased to inform you that your manuscript has been judged scientifically suitable for publication and will be formally accepted for publication once it meets all outstanding technical requirements.

Kind regards,

Immaculate Sabelile Tenza, PhD

Guest Editor

PLOS One
---

## [Editor Report · Acceptance letter]

PONE-D-25-55029R1

PLOS One

Dear Dr. Kawazoe,

I'm pleased to inform you that your manuscript has been deemed suitable for publication in PLOS One. Congratulations! Your manuscript is now being handed over to our production team.

Kind regards,

on behalf of

Dr. Immaculate Sabelile Tenza

Academic Editor

PLOS One